# Approaches to Identify Pregnancy Failure in Buffalo Cows

**DOI:** 10.3390/ani11020487

**Published:** 2021-02-12

**Authors:** Vittoria Lucia Barile, Laura Menchetti, Anna Beatrice Casano, Gabriele Brecchia, Noelita Melo de Sousa, Riccardo Zelli, Claudio Canali, Jean François Beckers, Olimpia Barbato

**Affiliations:** 1Council for Agricultural Research and Analysis of Agricultural Economics (CREA), Research Centre for Animal Production and Aquaculture, Monterotondo, 00015 Rome, Italy; vittorialucia.barile@crea.gov.it; 2Department of Agricultural and Food Sciences, University of Bologna, Viale G. Fanin 4, 40137 Bologna, Italy; laura.menchetti7@gmail.com; 3Department of Veterinary Medicine, University of Perugia, via San Costanzo, 06126 Perugia, Italy; annabeatrice.casano@libero.it (A.B.C.); riccardo.zelli@unipg.it (R.Z.); claudio.canali@unipg.it (C.C.); 4Department of Veterinary Medicine, University of Milano, via dell’Università, 26900 Lodi, Italy; gabriele.brecchia@unimi.it; 5Laboratory of Animal Endocrinology and Reproduction, Faculty of Veterinary Medicine, University of Liege, 4000 Liege, Belgium; noelitamelo@gmail.com (N.M.d.S.); jfbeckers@ulg.ac.be (J.F.B.)

**Keywords:** embryonic mortality, pregnancy-associated glycoproteins, progesterone, ultrasonography, buffalo

## Abstract

**Simple Summary:**

Embryonic mortality and pregnancy failures still represent a major issue in domestic livestock production, particularly in dairy cattle. Despite the presence of extensive work in this research area, there is still no effective, accurate and practical method able to determine timing and viability of embryo specifically during early gestation. Indeed, technologies and techniques for predicting pregnancy success must continue to be developed. The aim of this work was to find the best strategy to diagnose pregnancy failures in buffalo cows in order to improve farm reproductive management. Among the methods compared in this study (ultrasonography, progesterone, PAGs), pregnancy-associated glycoproteins (PAGs) seem to be the best marker for predicting embryonic mortality between 25 and 40 days of gestation to be utilized as a diagnostic tool to improve reproductive management in buffalo farms.

**Abstract:**

The aim of this work was to find the best strategy to diagnose pregnancy failures in buffalo. A total of 109 animals belonging to a buffalo herd subjected to a synchronization and artificial insemination (AI) program were enrolled in this study. Blood samples were collected at days 0, 14, 25, 28 and 40 after AI for the determination of progesterone (P4) and pregnancy-associated glycoproteins (PAGs) by the radioimmunoassay (RIA) method. Transrectal ultrasonography was performed on day 25, 28 and 40 after AI to monitor pregnancy. The animals included in the data analysis were assigned ex post in pregnant (*n* = 50) and mortality (*n* = 12) groups. By ultrasonography, the predictive sign of mortality was the heartbeat. At day 25, the PAGs concentration was significant in predicting embryonic mortality with respect to ultrasonography and P4, at the cut-off of 1.1 ng/mL. At day 28, either PAGs, at a cut-off of 2.2 ng/mL, or ultrasonography, with no detection of heartbeat, were highly predictive of embryonic mortality. PAGs were the best marker (*p* < 0.05) for predicting embryonic mortality between 25 and 40 days of gestation in buffalo. Its utilization as a diagnostic tool can influence management decisions in order to improve farm reproductive management.

## 1. Introduction

Animal reproductive biotechnology is continually evolving. Advances have been made in our understanding of embryo development and early embryonic mortality in domestic animals, which has improved the selection and success of in vivo and in vitro technologies. Declining fertility is a globally recognized problem that represents a major source of economic loss and culling in livestock species [1,2]. The low reproductive efficiency of cattle has had a negative financial impact on the dairy industry because the higher yields of milk cannot compensate for dwindling herd sizes. The demand for assisted reproductive techniques is therefore increasing, by creating an additional cost for dairy farmers [3]. Many factors contribute to the decline in reproductive fitness. The major cause of poor reproductive success is early embryonic mortality [4], which is defined as pregnancy failure during the period between fertilization and day 42 of gestation [5]. This is true especially in the animals that are not mated during their reproductive period.

Although buffaloes are polyestrus, their reproductive efficiency changes throughout the year, showing a distinct seasonal pattern [6]. This reproductive seasonality affects the efficiency of breeding programs, particularly during the spring–summer season with the daylight lengthening period, which corresponds to the low breeding period for buffalo [7]. It was observed a higher incidence of embryo loss (20 to 40%) in buffaloes that conceive during the daylight lengthening period, whereas a lower incidence (7%) was observed during decreasing daylight length [8,9,10,11]. Such high rates of early embryonic mortality lead to substantial losses in time and money spent rebreeding cows, slower genetic progress, as well as a significant loss in the number of potential calves [12,13]. One strategy for improving reproductive performance aims to shorten the calving-conception interval by rapidly identify embryo losses and rebreeding non-pregnant cows [14]. Pregnancy can be monitored using a variety of methods, including direct methods such as ultrasonography, or indirect methods like progesterone (P4) or pregnancy-associated glycoproteins (PAGs) measurement in maternal blood [15,16].

In buffalo, applications of transrectal ultrasonography to monitor early pregnancy and embryonic development have been described by different authors [17,18,19,20,21]. The sensitivity (true positive) of transrectal ultrasonography between days 19 and 24 is reported to be 44.4%, reaching 100% from day 31 after mating, while the specificity (true negative) ranged between 92.5 and 100% from days 19 to 55 after mating [22]. The time of heartbeat detection can be considered as the moment at which the sensitivity of the gestational diagnosis is 100% [20]. In buffalo, embryo proper with a heartbeat can be visualized between 23 and 28 days after gestation [20]. No detectable heartbeat together with membrane detachment/disorganization, reduced amount of intrauterine fluid or echoic floating structures, including remnants of the conceptus, are ultrasound findings of an embryo death [23]. Therefore, using ultrasonography, early pregnancy losses can be clearly diagnosed. The disadvantage of this method is that accuracy is limited before 28 to 30 days of gestation and pregnancy status is only guaranteed at the time of diagnosis; moreover, when the fetal heartbeat of an embryo is viewed on an ultrasound, there is no indication of whether or not embryo mortality will occur [16].

Chemical-based methods have been developed to identify pregnancy in buffalo, as an alternative to the ultrasound technique [24].

Progesterone (P4) is the most biologically active progestagen and is primarily produced and secreted by the corpus luteum (CL) during the estrous cycle and by the placenta during pregnancy. Quantification of P4 in blood or milk at days 20, 22 and 24 post-breeding has been utilized for early pregnancy diagnosis in buffaloes [25,26]. Campanile et al. [8] found higher P4 plasma levels in pregnant buffaloes than in buffaloes that showed embryonic mortality since day 10 after AI. Although P4 concentration helps in detecting early pregnancy, a single analysis does not provide sufficient information to evaluate the pregnancy status accurately [27,28]. Indeed, the concentration of P4 reflects the function of the CL and not the presence or the vitality of an embryo or foetus.

Characterized for the first time in the early eighties, the pregnancy-associated glycoproteins (PAGs) constitute a large family of glycoproteins expressed in the outer epithelial layer (chorion/trophectoderm) of the placenta in eutherian species [29,30,31]. They are synthesized by the mononucleate and binucleate trophoblastic cells, some of them being secreted into the maternal blood from the moment when the conceptus becomes more closely attached to the uterine wall and placentome formation begins [32,33]. Using different chromatographic procedures, some members of the PAGs family have been isolated from the cotyledons of different species of Cetartiodactyla order [29,34,35,36,37,38,39,40,41,42,43], included buffalo species [44,45]. The accumulation of PAGs in the maternal blood of ruminant ungulates has become a useful tool for monitoring pregnancy, thanks to the development of homologous [46] and heterologous radioimmunoassay (RIA). In our previous works, our research group [45,47,48,49] described the use of antisera raised against buffalo PAGs for RIA development and pregnancy detection in buffalo cows. Previously, other authors [22,50,51] used the RIA-706 system to measure PAGs in this species. Recently, the molecular biology technique has been utilized to detect the mRNA expression of PAGs in the maternal blood of pregnant buffaloes [52,53,54], improving knowledge regarding the peri-implantation period and the earliest time in which PAGs could be detected in this species.

The aim of this work was to find the best strategy to diagnose pregnancy failures in buffalo comparing ultrasonography, P4 and PAGs in order to improve farm reproductive management. Moreover, the work highlights whether PAGs determination could support the diagnosis of early pregnancy failures in buffalo.

## 2. Materials and Methods

### 2.1. Animals and Experimental Design

The trial was carried out at the CREA Animal Production and Aquaculture experimental farm of Monterotondo (Rome, Italy, 42° N parallel). A total of 109 animals belonging to the Italian Mediterranean buffalo herd subjected to a synchronization and artificial AI program were enrolled in this study and grouped as described below. Buffaloes were kept on a loose-housing system, fed ad libitum once a day with a total mixed ration based on sorghum silage, hay and concentrate and milked twice a day in a milking parlor. Before estrous synchronization, regular clinical examination excluded diseases such as endometritis, mastitis and metabolic disorders.

Buffaloes were synchronized with a progesterone releasing intravaginal device (PRID; Sanofi, Paris, France), containing 1.55 g natural progesterone inserted in situ for 10 days and an i.m. injection of 1000 IU of Pregnant Mare Serum Gonadotrophin (PMSG; Ciclogonina, Fort Dodge, Bologna, Italy) and 0.15 mg of cloprostenol (PGF2α analogue; Dalmazin, Ozzano Emilia, Fatro, Italy) on day seven. On day 10, the PRID was removed and cows were artificially inseminated at 72 and 96 h from device withdrawal.

Blood samples were taken from the jugular vein in 10 mL EDTA tubes at days 0 (0d), 25 (25d), 28 (28d) and 40 (40d) after AI for P4 and PAGs analysis. On day 14 (14d), a further sample was taken for P4 analysis. The day of the second AI was considered as day zero. Plasma was immediately separated by centrifugation (1200× *g* for 15 min at 5 °C) and stored at −20 °C until assayed.

The animals were classified ex post as pregnant (*n* = 50) and mortality (*n* = 12) groups, as determined by ultrasonography at day 40 based on diagnostic criteria as reported below in paragraph 2.2. A total of 47 buffaloes were diagnosed as non-pregnant. Only animals related to pregnant and mortality groups were included in the data analysis.

The animals involved in this experiment were treated in compliance with the animal testing regulations established under Italian law. The experimental design was carried out according to good veterinary practices under farm conditions. The CREA Research Centre for Animal Production and Aquaculture is authorized to use farm animals for experimental design (as stated in DM 26/96-4 of the Italian Welfare Ministry).

### 2.2. Pregnancy Diagnosis

Transrectal ultrasonography (Aloka SSD Prosound 2 scanner, Hitachi Medical System, Buccinasco, Italy, equipped with a 7.5 MHz linear-array transducer) was performed by the same operator on days 25 (25d), 28 (28d) and 40 (40d) after AI in correspondence with blood sampling for PAGs. On the day of scan, the ultrasound observations were classified into four categories: no vesicle, vesicle, vesicle + embryo, and vesicle + embryo + beat.

Recognition of the embryonic vesicle and embryo proper with a heartbeat was used as the criterion for positive diagnosis. Embryo mortality was diagnosed when a first embryo vesicle and/or embryo proper with a heartbeat was no longer visible in later ultrasound scans. Non-pregnant buffaloes had no embryonic vesicle detected at any time point. Pregnancy status was confirmed on day 60 post-AI.

### 2.3. Progesterone Radioimmunoassay

Samples were assayed for P4 using an extraction step as described previously [47,55,56]. P4 was extracted from plasma using ethyl ether. Each sample was assayed in duplicate. The efficiency of the extraction of a tracer amount of [3H]-progesterone ranged from 82 to 95%. Extraction was conducted using 0.2 mL plasma. Briefly, 0.8 mL distilled water and 3 mL diethyl ether were added to each sample. After stirring, samples were centrifuged (1000× *g*, 10 min), frozen, and the supernatant was discarded. Then, 1 mL borate buffer containing 10% ethanol was added to each sample. A 100-mL volume of increasing concentrations of P4 (0.1, 0.25, 0.5, 2, 5, 10, 20, and 40 ng/mL) constituted the standard curve. Tubes corresponding with the total count (Tc), nonspecific binding (NSB), and reference samples (0.5 and 10 ng/mL) were also prepared. Volumes of 100 mL [3H]-progesterone and 100 mL of diluted antiserum were added to the tubes containing the extracted samples and standard curve. Incubation was performed for at least 4 h at 4 **°**C. Bound and free P4 were separated by centrifugation after dextran-charcoal adsorption. Tubes were counted in a beta-counter (Tri-carb 2100 TR; Packard; Milan, Italy). The minimum detection limit (MDL) was 0.08 ng/mL. Intra-assay and interassay coefficients of variation (CV) were 7% and 11%, respectively.

Based on P4 assay, buffaloes were considered pregnant when the plasma P4 concentrations on days 14, 25, 28 and 40 were over 1 ng/mL.

### 2.4. PAGs Radioimmunoassay

For PAGs concentration, RIA-860 previously described by Barbato et al. [45] was used. Pure boPAG_67kDa_ preparation was used as the standard and tracer. Iodination (Na-I^125^, Amersham Pharmacia Biotech, Uppsala, Sweden) was carried out according to the chloramine-T method previously described by Greenwood et al. [57]. The samples were assayed in a preincubated system in which the standard curve ranged from 0.2 to 25 ng/mL.

The minimum detection limit (MDL), calculated as the mean concentration minus twice the standard deviation (mean–2 SD) of 20 duplicates of the zero (B_0_) standard [58], was 0.4 ng/mL. The intra- and inter-assay coefficients were 2.8% and 7.1%, respectively.

Based on PAGs assay, (cut-off value: ≥1 ng/mL) buffaloes were considered non-pregnant when concentrations remained very close to zero at all time points and pregnant when concentrations were higher than 1 ng/mL at days 25, 28 and 40. When PAGs concentrations were ≥1 ng/mL at day 25 and dropped under 0.2 ng/mL by day 40, embryo mortality was considered to have occurred.

### 2.5. Statistical Analysis

The association between the outcome (2 levels: Pregnant and Mortality) and results of the ultrasound analysis (4 levels: No vesicle, Vesicle, Vesicle + embryo, Vesicle + embryo + beat) was analyzed by Chi-square or Fisher’s exact test, stratifying for day of observation (time). Z-tests were used to compare column proportions. This association was not evaluated at time 0 as all the animals were in the no vesicle group.

The changes of P4 and PAGs concentrations with time in the two groups were instead analyzed using linear mixed models (LMMs). In LMMs, animals and days were included as subjects and repeated factors, respectively. The LMMs evaluated the main effects of time (4 levels: 0, 25, 28, and 40 days post-AI), outcome (2 levels: pregnant and mortality), and the interaction between the outcome and time. Sidak adjustment was used for carrying out multiple comparisons. Diagnostic graphics were used for testing assumptions and outliers. Because non-normality of the data was detected for P4 and PAGs concentrations, log and log(x + 1) transformation, respectively, were used for analysis. Back-transformed estimated marginal means were presented as results while the row data were presented as figures. Moreover, the association between P4 and PAGs concentrations was analysed using the Spearman’s correlation coefficient (ρ).

In order to test the ability of PAGs and P4 concentrations to discriminate between mortality and pregnant outcomes at each time (days 25, 28, and 40 post-AI), the receiver operating characteristic (ROC) analysis was also performed and the optimal cut-off was determined by Youden index [59]. Finally, univariate models were built using the generalized linear models procedure with a binomial distribution and the logit link function to evaluate the accuracy of the parameters in predicting mortality at day 25 and 28. PAGs and P4 concentrations were categorized according to the cut-off for each day while ultrasound outcome was categorized according to the identification of the embryo heartbeat. Odds ratios (ORs) with the corresponding 95% confidence interval (CI) and p-values were calculated.

Statistical analyses were performed with SPSS Statistics version 25 (IBM, SPSS Inc., Chicago, IL, USA). Statistical significance occurred when *p* ≤ 0.05.

## 3. Results

A total of 50 out of 109 buffalo cows enrolled in this study became pregnant (pregnant group) while 12 had embryo mortality (mortality group) and 47 remained non-pregnant as determined by ultrasonography at day 40. All buffaloes diagnosed as pregnant were confirmed at day 60.

### 3.1. Ultrasound Observations and Embryo Mortality

Results of ultrasound observations according to the outcomes (pregnant vs. mortality) are summarized in Table 1. At day 25, the proportion of animals in which the vesicle was not identified was greater in the mortality group, while the proportion of animals in which the vesicle, vesicle + embryo or vesicle + embryo + beat was identified were higher in the pregnant group (*p* < 0.0001). At day 28, the proportion of animals in which the vesicle + embryo was identified was higher in the mortality group while the animals in which vesicle + embryo + beat were identified was higher in pregnant buffaloes than those in the mortality group (*p* < 0.0001). A significant association between the ultrasound observations and the outcome was also found at day 40; vesicle + embryo + beat was identified in all the buffaloes in the pregnant group; in the mortality group, 11 buffalo cows (91.7%) showed only the vesicle and 1 cow (8.3%) the vesicle + embryo (*p* < 0.0001).

### 3.2. P4 Concentrations and Mortality

P4 concentrations were affected by time (*p* < 0.001), outcome (*p* < 0.001), and interaction (*p* < 0.001; Figure 1). Significant differences in P4 concentrations between the mortality and pregnant groups were found at days 25 (mean difference: 1.8 ± 0.1 ng/mL, *p* < 0.001), 28 (mean difference: 1.4 ± 0.1 ng/mL, *p* < 0.05) and 40 (mean difference: 3.1 ± 0.1 ng/mL, *p* < 0.001) post-AI.

The ROC curves for detection of embryo mortality by P4 and the optimal cut-off for predicting mortality are reported in Table 2 and Figure 2.

ROC analysis for P4 concentrations at day 25 showed an area of 0.793 (Figure 2, Panel A) and the optimal cut-off for predicting mortality of 2.6 ng/mL. A sensitivity of 100% was achieved for P4 concentrations lower than 4.1 ng/mL. At day 28, the AUC was 0.722 (Figure 2, Panel B) and the optimal cut-off was 2.6 ng/mL. At day 40, the AUC was 0.883 (Figure 2, Panel C) and the optimal cut-off was 2.4 ng/mL. A sensitivity of 100% at day 40 post-AI was achieved for P4 concentrations lower than 4.2 ng/mL.

### 3.3. PAGs Concentrations and Mortality

Significant effects of time (*p* < 0.001), outcome (*p* < 0.001) and interaction were observed in the log-PAG-1 concentrations (*p* < 0.001). Differences in PAGs levels between the two groups started from day 25 post-AI (mean difference: 0.3 ± 0.1 ng/mL, 0.3 ± 0.1 ng/mL, and 11.6 ± 0.1 ng/mL at days 25, 28, and 40 post-AI, respectively; *p* < 0.01; Figure 3).

The results of the ROC analyses performed at each day are reported in Table 2 and Figure 4.

At day 25, the area under ROC was 0.837 and the Youden Index analysis revealed that the optimal cut-off for predicting mortality was 1.1 ng/mL (Figure 4, Panel A). A sensitivity of 100% was achieved for PAGs concentrations lower than 1.6 ng/mL. At day 28, the AUC was lower (0.700) and increased the optimal cut-off value (2.2 ng/mL; Figure 4, Panel B). At day 40, PAGs concentration perfectly discriminated between the mortality and pregnant groups (AUC = 1.000; Figure 4, Panel C) and the cut-off of 2.7 ng/mL identified cases of mortality with a sensitivity of 100% and a specificity of 100%.

### 3.4. Association between Ultrasound Outcome and PAGs Concentrations

In the model including days 25 and 28 as time and the classification based on the ultrasound observations as fixed effect, we found a significant effect of time (*p* < 0.05), group (*p* < 0.001), and interaction (*p* < 0.05) on PAGs concentrations.

There were no differences on PAGs concentrations at day 25 according to ultrasound observations. At day 28, Vesicle (1.5 ± 0.1 ng/mL; *p* < 0.01) and Vesicle + embryo (1.5 ± 0.1 ng/mL; *p* < 0.05) had lower PAGs concentrations compared with Vesicle + embryo + beat (3.3 ± 0.1 ng/mL).

### 3.5. Univariate Models Determining Predictors of Mortality at Days 25 and 28 Post-AI

The lack of embryo heartbeat detected by ultrasound at day 25 was not a predictor of embryonic mortality (*p* < 0.119) while at day 28 the odds of mortality increased by about 15 times if the heartbeat was not detected (Table 3).

Conversely, P4 and PAGs were predictors of mortality both at day 25 and 28 post-AI (Table 3). Both at days 25 and 28, the odds of mortality were six times higher when PAGs concentrations were lower than or equal to the cut-off (1.1 ng/mL and 2.2 at days 25 and 28, respectively; *p* < 0.05) and five times higher if P4 was lower than or equal to 2.6 ng/mL (*p* < 0.05).

## 4. Discussions

The aim of this work was to find the best strategy to diagnose pregnancy failures in buffalo cows in order to improve farm reproductive management through intervention strategies for animals identified to be at risk for embryo loss. From the results of ultrasound at day 25, it seems that the animals that experienced embryonic mortality had a delayed growth of the vesicle and embryo since the vesicle was not observed in more than half of the buffalo cows in the mortality group, while those that maintained pregnancy already showed vesicle + embryo + beat in a higher percentage at the same time. Confirming this hypothesis, the higher rate of vesicle + embryo + beat in the animals that later experienced embryonic mortality was found at day 28.

Uterine environment is critical as the embryo transcends from the oviduct to the site of implantation. Asynchrony between the uterus and the embryo can be problematic as the uterine environment will not wait for an embryo, although an embryo can accelerate or decelerate its development to some degree [60,61]. At day 40, all the animals in which there was a recognized embryonic vesicle and the embryo proper with heartbeat maintained pregnancy until the end of the observation period (60 days post-AI), while no buffaloes in the mortality group showed vesicle + embryo + heartbeat. Pregnancy diagnosis is considered positive only when gestational vesicle and embryo, together with a heartbeat, can be detected [62]. In bovine pregnancy, based on the absence of a heartbeat, placental detachment, or reduced placental fluid volume, pregnancy losses have been diagnosed and reported to occur between 24 and 40 days of gestation [63,64,65]. In the present study, using the ultrasound observations, most of embryonic mortality was diagnosed as occurring between 28 and 40 days after insemination as reported by other authors in this species [8,66,67].

Embryo death can be clearly diagnosed by an undetectable heartbeat. In fact, at day 28 post-AI, the odds of mortality increased by about 15 times if the heartbeat was not detected. The disadvantage of this method is that before 28 days of gestation it is not predictive of embryonic mortality. This is in agreement with what has been reported by other authors of bovine pregnancy studies [16,60,68].

Regarding the biological markers of pregnancy, buffaloes that maintained pregnancy had significantly higher circulating concentrations either of PAGs or P4 starting from day 25 after AI until the end of the observation period (40 days after AI).

Progesterone is primarily produced and secreted by the CL and subsequently by the placenta during pregnancy [69]; thus, the concentration of progesterone in the first weeks of pregnancy reflects the function of the CL more than the presence of an embryo. The persistence of the CL after an embryonic death or the start of a new oestrus cycle after embryo resorption leads to a positive value of P4 in the bloodstream [24]; therefore, the presence of P4 that we found in the mortality group at 40 days post-AI, when embryo resorptions were recorded by ultrasound observation, could be due to this reason.

Starting from day 25 post-AI, significant differences in the P4 value were found between animals that experienced embryo mortality and those that maintained pregnancy. The lower P4 value in the mortality group could indicate a lower functionality of the CL that could have contributed to the embryo resorption [69,70]. In a previous work, the diameter and echogenicity of the CL seemed to affect its functionality as there was a positive correlation between plasma P4 concentration and CL diameter that was found to be significantly larger in pregnant buffalo in contrast to non-pregnant buffalo [71]. Campanile et al. [72] reported that buffaloes that underwent late embryonic development had relatively low concentrations of P4 in their blood; this finding was interpreted to indicate that reduced P4 concentrations were inadequate to induce changes in the uterus required for attachment of the conceptus.

Although P4 concentration helps in detecting early pregnancy, it is not highly specific and has been the major problem for the prevalence of false-positives (identifying a non-pregnant animal as pregnant) [16,73]; moreover, P4 is not useful for verifying the presence of a viable embryo in the uterus [74]. Therefore, the non-specificity of P4 and the presence of possible false positives has limited the use of this method as a marker of embryonic viability [28,75,76].

Different from P4, PAGs indicate the presence of an embryo in the uterus of eutherian species. Several authors showed the relationship between PAGs and fetal wellbeing [77,78,79,80,81]. These glycoproteins are synthesized by the giant trophoblast cells (TGCs) which migrate from the fetal to the uterine epithelium and release PAGs into the maternal blood [32,33,82]. This process presumes the presence of a healthy trophoblastic tissue and thus of a healthy embryo. If this condition fails, the source of production of the proteins is missing. Furthermore, thanks to this “active migration”, PAGs play an important role in the remodeling of fetal membranes and in the formation of placentome during pregnancy [81,83,84], and is a possible factor controlling maternal immune modulation [85,86]. Therefore, in addition to serving as an accurate tool for diagnosing pregnancy in ruminants, PAGs may also serve as a marker for monitoring embryonic/fetal viability and placental function [16,60,87,88,89]. In buffalo, this protein can be detected in the blood of pregnant animals starting from day 25 post-AI, with an accuracy of 99% on day 28 of gestation, thus providing an accurate test to follow up pregnancy [45,49,90].

In this work, buffaloes that experienced embryo mortality had a lower concentration of PAGs starting from day 25 post-AI, suggesting a possible embryo suffering in these animals at the early stage of gestation. In cows, physiologically, pregnancy loss after day 28 of gestation coincides with a period of active placentation marked by extensive endometrium remodeling, binucleate cell migration and changes in PAGs expression production [68].

On the basis of the ROC curves and positive and negative analysis, an optimal PAGs cut value has been established that is 74% accurate in predicting early embryonic mortality for a concentration less than 1.1 and 2.2 ng/mL at day 25 and 28 post-AI, respectively. To our knowledge, no data are reported in the literature regarding a cut-off predicting embryonic mortality in buffalo. In bovine pregnancy, Polher et al. [16] suggest that PAGs are predictive of embryonic mortality between day 28 and 45 of gestation. The same authors in a later work [91] have shown that at day 28 of gestation a circulating concentration of PAGs greater than 7.9 ng/mL was 95% accurate in predicting embryonic maintenance (to day 100), and a concentration of glycoproteins less than 0.72 ng/mL was 95% accurate in predicting embryonic mortality by day 100. Gatea et al., [92] reported that circulating concentrations of PAGs on days 28 to 31 have been shown to be measurably lower in cows that experienced late embryonic mortality.

Based on odds ratio analysis, our data showed that for buffalo cows with PAGs values ≤1.1 and ≤28 ng/mL at day 25 and 28 post-AI, respectively, there was a 6.3 times greater chance to undergo pregnancy loss within the first 40 days of gestation compared to those that had higher values. Higher PAGs concentrations were correlated to a decreased incidence of embryonic loss as reported also in other bovine studies [93,94].

In a study on the accuracy of ultrasonography and PAGs for pregnancy diagnosis in buffaloes, Karen et al. [22] reported no significant differences between the sensitivity and specificity of the two tests in the examined period (19 and 55 days after mating), and concluded that both are highly accurate tests for detecting pregnant buffaloes from day 31 onwards after mating. In this study, the ultrasound allowed us to significantly detect the presence of a viable embryo (vesicle + embryo + heartbeat) at day 28 post-AI, predicting the animals that would maintain pregnancy compared to those that would experience embryonic mortality, while PAGs concentrations permitted the discrimination between buffalo that experienced embryonic mortality and those that maintained pregnancy starting from 25 days of gestation.

## 5. Conclusions

Among the methods investigated in this study, PAGs were the best marker for predicting embryonic mortality in buffalo between 25 and 40 days of gestation. Although low values of P4 were associated with pregnancy failures, as a predictor for pregnancy loss, P4 is less reliable compared to PAGs and ultrasonography. The disadvantage of ultrasonography is that the pregnancy status is only guaranteed at the time of diagnosis. Differently, PAGs reflect embryo wellbeing and therefore the reduction of its circulating concentrations is a prognostic sign of pregnancy failure. Notwithstanding that the assay refinement and studies with a large sample size are needed to improve the predictive value to an acceptable point for use in applied reproductive management, our data have shown that PAGs could be utilized as a diagnostic tool in order to improve farm reproductive management through intervention strategies for animals identified to be at risk for embryo loss.

## Figures and Tables

**Figure 1 animals-11-00487-f001:**
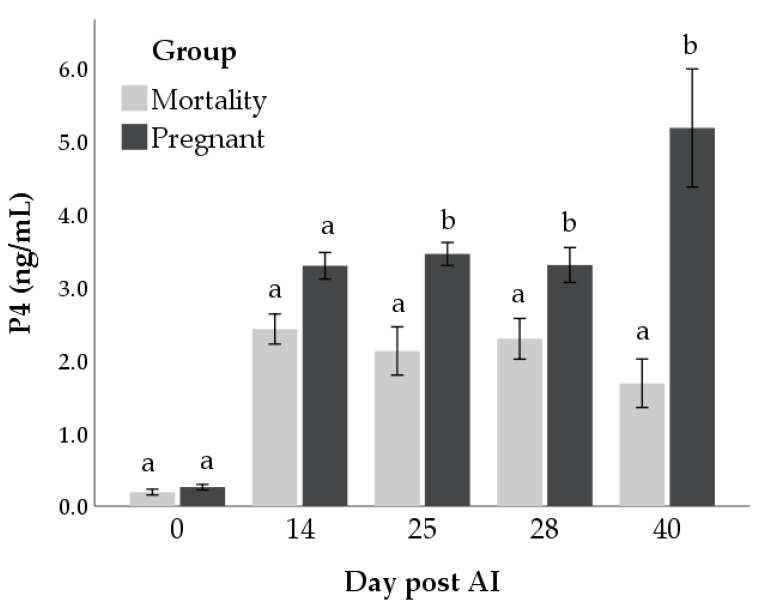
Concentrations of progesterone (P4) in mortality and pregnant groups at day 0, 14, 25, 28 and 40 post-AI. Bars not sharing the same superscript within each day are significantly different at *p* < 0.05.

**Figure 2 animals-11-00487-f002:**
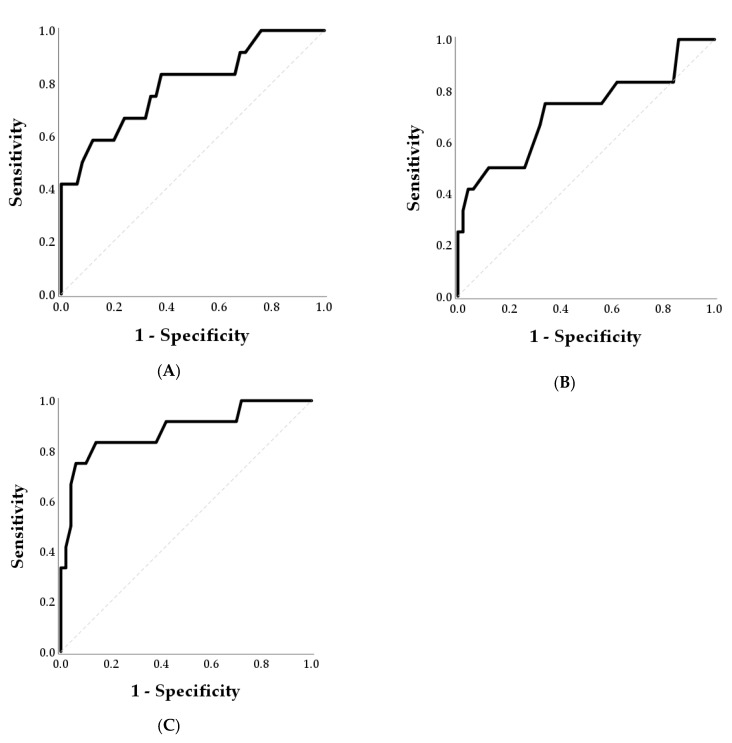
Receiver operating characteristics curves for the detection of embryonic mortality by P4 at days 25 (Panel **A**), 28 (Panel **B**), and 40 (Panel **C**) post-AI. Optimal cut-offs for predicting mortality were 2.6 ng/mL, 2.6 ng/mL, and 2.4 ng/mL at days 25, 28 and 40 post-AI, respectively.

**Figure 3 animals-11-00487-f003:**
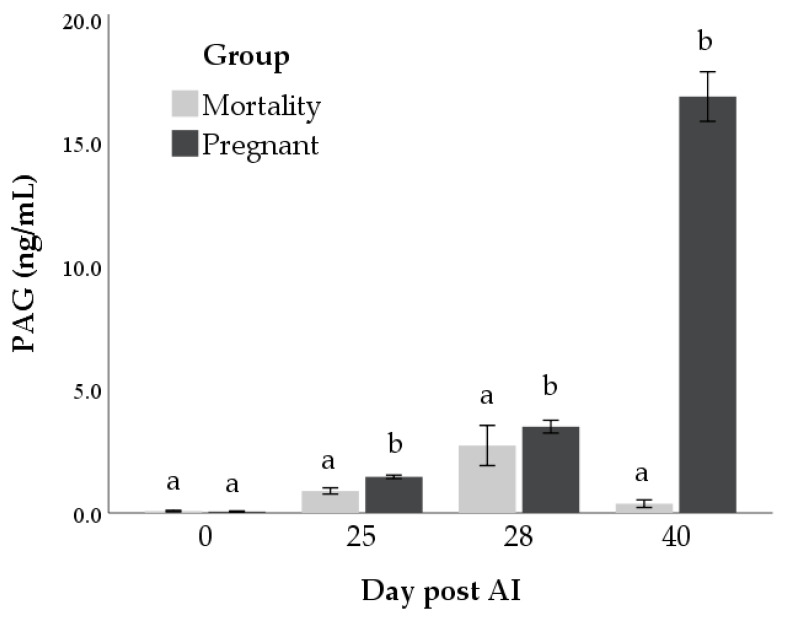
Concentrations of pregnancy-associated glycoproteins (PAG) in mortality and pregnant groups at days 0, 25, 28, and 40 post-AI. Bars not sharing the same superscript within each day are significantly different at *p* < 0.05.

**Figure 4 animals-11-00487-f004:**
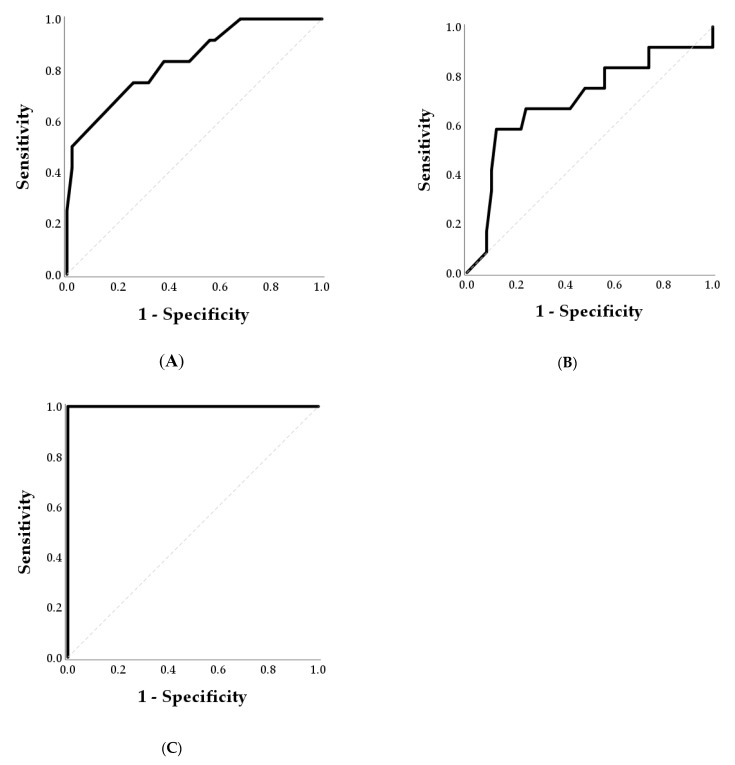
Receiver operating characteristic (ROC) curves for the detection of embryonic mortality by PAG at days 25 (Panel **A**), 28 (Panel **B**), and 40 (Panel **C**) post-AI. Optimal cut-offs for predicting mortality were 1.1 ng/mL, 2.2 ng/mL, and 2.7 ng/mL at days 25, 28 and 40 post-AI, respectively.

**Table 1 animals-11-00487-t001:** Results of ultrasound observations according to number of days post-artificial insemination (AI) and outcome.

Day Post-AI	Ultrasound Observation	Outcome		Significance
		Mortality	Pregnant	
25	No vesicle	8 ^a^ (66.7%)	4 ^b^ (8.0%)	0.0001
	Vesicle	1 ^a^ (8.3%)	22 ^b^ (44.0%)	
	Vesicle + embryo	1 ^a^ (8.3%)	3 ^a^ (6.0%)	
	Vesicle + embryo + beat	2 ^a^ (16.7%)	21 ^b^ (42.0%)	
28	No vesicle	1 ^a^ (8.3%)	0 ^a^ (0.0%)	0.0001
	Vesicle	2 ^a^ (16.7%)	3 ^a^ (6.0%)	
	Vesicle + embryo	3 ^b^ (25.0%)	0 ^a^ (0.0%)	
	Vesicle + embryo + beat	6 ^a^ (50.0%)	47 ^b^ (94.0%)	
40	No vesicle	11 ^b^ (91.7%)	0 ^a^ (0.0%)	0.0001
	Vesicle + embryo	1 ^a^ (8.3%)	0 ^a^ (0.0%)	
	Vesicle + embryo + beat	0 ^a^ (0.0%)	50 ^b^ (100.0%)	

Values in the same row not sharing the same superscript (a, b) are significantly different at *p* < 0.05 (z-test).

**Table 2 animals-11-00487-t002:** Results of Receiver Operator Characteristic (ROC) analysis including the area under the ROC curve (AUC) with corresponding 95% confidence intervals (CI) and *p*-values, optimal cut-off with associated sensitivity, specificity and accuracy values.

Day	Parameter	AUC	95% CI	*p*-Value	Cut-off (ng/mL)	Sensitivity (%)	Specificity (%)	Accuracy (%)
25	PAG	0.837	0.706–0.967	<0.001	1.1	75	74	74
	P4	0.793	0.639–0.946	0.002	2.6	67	76	74
28	PAG	0.700	0.516–0.884	0.033	2.2	67	76	74
	P4	0.722	0.537–0.906	0.018	2.6	75	66	68
40	PAG	1.000	1.000–1.000	<0.001	2.7	100	100	100
	P4	0.883	0.758–1000	<0.001	2.4	83	86	85

PAG = Pregnancy-Associated Glycoproteins. P4 = Progesterone.

**Table 3 animals-11-00487-t003:** Univariate models to evaluate the predictors of mortality in buffalo cows at days 25 and 28 post-AI.

Day Post-AI	Predictor	OR			95% CI		*p*-Value
					Lower	Upper	
**25**	**Ultrasound outcome**						
	No embryo + beat vs. Embryo + beat	3.621			0.717	18.272	0.119
	**PAG concentrations** ≤1.1 ng/mL vs. >1.1 ng/mL	6.375			1.517	26.784	0.011
	**P4 concentration** ≤2.6 ng/mL vs. >2.6 ng/mL	4.667			1.217	17.894	0.025
**28**	**Ultrasound outcome**						
	No Embryo + beat vs. Embryo + beat	15.667			3.083	79.613	0.001
	**PAG concentrations** ≤2.2 ng/mL vs. >2.2 ng/mL	6.333			1.618	24.786	0.008
	**P4 concentrations** ≤2.6 ng/mL vs. >2.6 ng/mL	4.895			1.176	20.372	0.029

Dependent variable: mortality. OR = odds ratio. CI = Wald confidence interval.

## Data Availability

No new data were created or analyzed in this study. Data sharing is not applicable to this article.

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
