# Peer review of "Approaches to Identify Pregnancy Failure in Buffalo Cows"

_animals, 2021, doi:10.3390/ani11020487_

Round 1

Reviewer 1 Report

The contribution of this study is marginal.

The study has the only merit to make a comparison among three different diagnostic tools to confirm the presence or loss of pregnancies in buffaloes.

Reproductive management of any farm dealing with animal production has to take into account not only the most effective strategy in terms of success or failure, but also the economic burden weighing on the farmer when deciding to adopt one or the other strategy. In other terms, authors should have made a comparison also in terms of costs among the three different diagnostic approaches: 1) cost of P4 analysis, 2) cost of PGAs analysis, and 3) cost of ultrasound.

In "Introduction" authors have rightly highlighted an important aspect related to "....reproductive efficiency and profitability on dairy farms...". This is evidently the same goal to be pursued in the dairy buffalo industry. Therefore, both efficiency and costs of any approach has to be considered.

As stated at the end in "Discussion", some authors have previously reported no difference in diagnostic efficiency and sensitivity between ultrasound and PGAs analysis in the period examined (Karen et al., 2007). Furthermore, in this study it is stated that the difference between ultrasound and PGAs analysis approaches, in terms of effective prediction of pregnancy maintenance or loss, is 3 days, starting at 25 days for PGAs analysis and 28 days for ultrasound diagnosis. If PGAs analysis costs should be confirmed higher than an ordinary ultrasound examination (ultrasound diagnosis in buffalo reproduction has become an ordinary tool these days), would 3 days difference in favor of PGAs analysis justify the adoption of his last diagnostic tool? Enrolling a non pregnant buffalo in a new synchronization protocol for AI 3 days later, is such a detrimental economic strategy for the farmer, when compared to enrolling the same animal 3 days earlier (ultrasound diagnosis vs PGAs analysis)? Of course, only one animal would not suffice to make a meaningful economic difference. Then, how many non pregnant buffaloes would make an economic difference if found 3 days earlier and enrolled immediately into a new synchronization protocol? This is a meaningful comparison, taking into consideration not only the efficiency and sensitivity of the methods, but also the costs involved.

Finally, authors should significantly improve the written English.

Author Response

Reviewer #1

R=Reviewer comment

A=Authors response

R: Reproductive management of any farm dealing with animal production has to take into account not only the most effective strategy in terms of success or failure, but also the economic burden weighing on the farmer when deciding to adopt one or the other strategy. In other terms, authors should have made a comparison also in terms of costs among the three different diagnostic approaches: 1) cost of P4 analysis, 2) cost of PGAsanalysis, and 3) cost of ultrasound”

A: To our knowledge, a comparison of ultrasound, P4, and PAGs to follow up embryo viability in buffalo was not reported before. So, with this work the authors wanted to give a contribution from the scientific point of wiew on this issue. The aim of work was to find the best strategy in term of efficiency and not in term of economic costs to diagnose pregnancy failures in buffalo comparing ultrasonography, P4 and PAGs. The results of the work suggest which test could be more predictive of embryonic mortality and could give an additional tool to people involved in the control of the reproductive management of a herd. Moreove,the choice of a method is dependent on what fits into the farm’s production system.

R: In "Introduction" authors have rightly highlighted an important aspect related to "....reproductive efficiency and profitability on dairy farms...". This is evidently the same goal to be pursued in the dairy buffalo industry. Therefore, both efficiency and costs of any approach has to be considered.

A: We reiterate that our work aimed to the efficiency and not to the affordability of the methods.

R: As stated at the end in "Discussion", some authors have previously reported no difference in diagnostic efficiency and sensitivity between ultrasound and PGAs analysis in the period examined (Karen et al., 2007).

A: Comparative studies of different methods to predict embryonic mortality are limited in buffalo. We have reported what we found on PAGs, ultrasound and / or P4 in the literature. In the case of Karen et al. (2017) the work referred to a comparison to evaluate the accuracy for the diagnosis of pregnancy (as we have reported), so it was not referred to embryonic mortality.

R: Furthermore, in this study it is stated that the difference between ultrasound and PGAs analysis approaches, in terms of effective prediction of pregnancy maintenance or loss, is 3days, starting at 25 days for PGAs analysis and 28 days for ultrasound diagnosis. If PAGs analysis costs should be confirmed higher than an ordinary ultrasound examination (ultrasound diagnosis in buffalo reproduction has become an ordinary tool these days), would 3 days difference in favor of PGAs analysis justify the adoption of his last diagnostic tool?

A: To our knowledge, the cost of PAGs is not higher than an ultrasound examination. The advantage of the PAGs is that having established, with this work , a cut off value predictive of embryonic mortality or survivability in buffalo, it could be used to identify animals that will not maintain pregnancy . From a pratical point of view, this could help the farm veterinarian in the choice of supporting pregnancy with an hormonal therapy in this critical period for the placentation in the animals showing embryo soffering. Moreover PAGs could be utilized also as a tool to identify non pregnant animals to enroller in new synchronization protocol and reserve the control of pregnancy with ultrasound of the animals PAGs positive later then 28gg to confirm pregnancy. Finally PAG could help in investigations related to the mechanisms involved in embryonic mortality in this species.

In the n order to highlight the scientific value of the research, respect to pratical implications, we have removed fom the manuscript the following sentence “allowing earlier resynchronization and rebreeding of the animals experienced pregnancy loss”

In any case we had already specified in the conclusions that “the assay refinement and studies with large sample size are needed to improve the predictive value to an acceptable point for use in applied reproductive management”.

Reviewer 2 Report

Dear author 

Attached you will find the manuscript containing my comments.

Author Response

Reply to reviewer #2

We thank the reviewer for suggestions that have been accepted and integrated in the revised manuscript except for the term referred to embryo loss/mortality(experienced/experiencing) because it is usaly utilized in this context. Below some exemples:

  • …..reported cows that experienced..( Pholer 2020)
  • Animals that experience early embryonic loss…..(Janna Kincheloe, Area Extension Specialist, Livestock Systems, NDSU, 2020)
  • …..reduced likelihood of conception…. (i.e., cows experiencing a disease with a hazard ratio ...(Lucy 2001)
  • …Whether cows calved or experienced pregnancy loss after d 76 was recorded…., (Bonilla ‎2014)

Reviewer 3 Report

Revision of the article entitled ‘Approaches to identify pregnancy failure in buffalo cows’ by Lucia Barile et al, Animals MDPI

This

Please check carefully guide for authors because there are some mistakes in the edition of the article

Abstract: can you change ‘it seems’ for a more scientific sentence (including significance P)

Introduction: Although it is well written, some paragraphs are too much redundant and disconnected. Please try to shorten this section and do not repeat ideas. For example, paragraph 3 should be explained in few lines, as well as 5 and 7. On the other size, you have paragraphs of 3 lines…is that a real paragraph? Can you include this sentence in other paragraph and try to connect ideas?

Material and methods:

Please, describe better the experimental farm apart from the number of animals

Ultrasonography: other criteria should be added to ensure you are not underestimating pregnancy losses by ultrasound: first the CL, 2 the heartbeat, 3 membrane detachment, 4 anechoic fluid (López-Gatius and Garcia-Ispierto, 2010; Ealy and Seekford, 2019).

The first paragraph of the statistical analyses is quite confusing. With a study population as low as this study, you cannot include 4 groups. What about if you analyze only dead or alive? All the criteria studied in the ultrasound examination should be focuses in determine if the embryo is dead or not.

Results:

It’s not clear for me, how do you know if a cow is pregnant or not at day 25. Of course, if it is prengnat on day 28 we know that the cow was pregnant. But what about negative ones? It’s quite confusing. Can you clarify that?

Discussion:

Did you expect your results? At least in cow, 25 days heartbeat is sometimes not visible, although the embryo is alive and detectable by ultrasound (Ginther, 1998). PAG is better than progesterone for detecting PL in cows too.

Please, correct that sentence ‘Embryo death can be clearly diagnosed by undetectable earth beat’

Be careful because there are some paragraphs, again of 3 lines, that are repeated from the results section.

This sentence of the concluding remarks are not really conclusions ‘From the comparison of investigated methods, PAGs seems to be the best marker for predicting embryonic mortality between day 25 and 40 of gestation in buffalo. Its utilization as diagnostic tool can influence management decisions in order to improve farm reproductive management, allowing earlier resynchronization and rebreeding of the animals experienced pregnancy loss. Moreover, PAGs could help in investigations related to the mechanisms involved in embryonic mortality in this species’

And one more question: are you taken into account twins? I know that in buffaloes twining pregnancies are scarce in literature, but with this well performed study in buffaloes you have done, you had the possibility of reporting cases. It’s just an idea.

Author Response

I thank the Reviewer 3 for the suggestions

Round 2

Reviewer 1 Report

As it is, without the additional information requested by this reviewer, this study remains of limited value.

Author Response

I thank the Reviewer 1 for the suggestions

Reviewer 2 Report

Thank you for accepting to follow my comments

Author Response

I thank the Reviewer 2 

Reviewer 3 Report

Dear authors

The paper has improved a lot. But I asked you a question: If you use ultrasound at day 40 to determine pregnancy as the gold standard technique, and you conclude that PAG are the best markers for pregnancy loss…don’t you think that maybe the ‘problem’ with this gestation is the corpus luteum? Can we do something with this cows? It’s just a reflexion. Can you add please a little sentence in the discussion section?

Thanks

Author Response

In the discussion section we have already told about CL in relation to P4 results:

“Starting from day 25 post AI, significantly differences in P4 value were found between animals that experienced embryo mortality and those that maintained pregnancy. The lower P4 value in Mortality group could indicate a lower functionality of CL that could have contributed to the embryo resorption [69, 70]. In a previous work, the diameter and the echogenicity of CL seemed to affect its functionality as there was a positive correlation between plasma P4 concentration and CL diameter, that was found to be significantively larger in the pregnant buffalo in contrast  to the non-pregnant [71]. Campanile et al., [72] reported that buffaloes that underwent late embryonic development had relatively low concentrations of P4 in blood and this finding was interpreted to indicate that reduced P4 concentrations were inadequate to induce changes in the uterus required for attachment of the conceptus.

So we think we have responded to the reviewer's request.